# Health Needs and Their Relationship with Life Expectancy in People with and without Intellectual Disabilities in England

**DOI:** 10.3390/ijerph19116602

**Published:** 2022-05-28

**Authors:** Freya Tyrer, Richard Morriss, Reza Kiani, Satheesh K. Gangadharan, Harish Kundaje, Mark J. Rutherford

**Affiliations:** 1Biostatistics Research Group, Department of Health Sciences, University of Leicester, Leicester LE1 7RH, UK; mjr40@le.ac.uk; 2Institute of Mental Health, University of Nottingham, Nottingham NG7 2TU, UK; richard.morriss@nottingham.ac.uk; 3Leicestershire Learning Disability Services (Psychiatry), Leicestershire Partnership NHS Trust, Leicester LE4 8PQ, UK; reza.kiani@nhs.net (R.K.); s.gangadharan1@nhs.net (S.K.G.); 4Mental Health, Ageing, Public Health and Primary Care Research Group, Department of Health Sciences, University of Leicester, Leicester LE1 7RH, UK; 5Lakeside Healthcare, Cottingham Road, Corby NN17 2UR, UK; harish.kundaje@nhs.net

**Keywords:** intellectual disability, life expectancy, health needs, epilepsy, incontinence, visual, hearing, mobility, cerebral palsy, PEG feeding

## Abstract

Health needs are common in people living with intellectual disabilities, but we do not know how they contribute to life expectancy. We used the Clinical Practice Research Datalink (CPRD) linked with hospital/mortality data in England (2017–2019) to explore life expectancy among people with or without intellectual disabilities, indicated by the presence or absence, respectively, of: epilepsy; incontinence; severe visual loss; severe visual impairment; severe mobility difficulties; cerebral palsy and PEG feeding. Life expectancy and 95% confidence intervals were compared using flexible parametric methods. At baseline, 46.4% (total *n* = 7794) of individuals with intellectual disabilities compared with 9.7% (total *n* = 176,807) in the comparison group had ≥1 health need. Epilepsy was the most common health need (18.7% vs. 1.1%). All health needs except hearing impairment were associated with shorter life expectancy: PEG feeding and mobility difficulties were associated with the greatest loss in life years (65–68% and 41–44%, respectively). Differential life expectancy attenuated but remained (≈12% life years lost) even after restricting the population to those without health needs (additional years expected to live at 10 years: 65.5 [60.3, 71.1] vs. 74.3 [73.8, 74.7]). We conclude that health needs play a significant role but do not explain all of the differential life expectancy experienced by people with intellectual disabilities.

## 1. Introduction

Addressing the burden of health inequalities is now a global priority [1,2,3]. Strategies to reduce these inequalities tend to focus on the most vulnerable, such as people living with disabilities or in areas of social deprivation [4,5,6,7,8]. Particularly at risk are those with intellectual disabilities (also known as learning disabilities in the UK) owing to a combination of genetic, social and behavioural factors [9,10]. Whilst there are measures in place to reduce health inequalities in this population, such as annual health checks [11], mortality data suggest that the situation has not improved, despite some deaths being potentially avoidable [12,13,14].

One of the challenges to reducing inequalities among people living with intellectual disabilities is that they are more likely than the general population to have severe health needs, including epilepsy, cerebral palsy and eating/feeding difficulties, which are known to shorten life expectancy [15]. Although not always life-limiting if managed well, they are relatively rare in the general population and so tend not to feature in population-level policy initiatives. Thus far, their individual contribution to life expectancy has not been formally investigated, but it is important to do so because this contribution may be over-inflated or seen as an inevitable consequence of having intellectual disabilities without seeking to improve health outcomes and/or quality of life for the individuals affected.

The aim of the current study was to investigate specific health needs and quantify their contribution to life expectancy in people with intellectual disabilities and to compare these findings with a cohort of individuals without intellectual disabilities. A further aim was to investigate people without any of the specified health needs to determine if loss in life years for people with intellectual disabilities remained.

## 2. Materials and Methods

### 2.1. Data Sources

This study followed the Reporting of studies Conducted using Observational Routinely-collected health Data (RECORD) checklist [16] (see Appendix A). We used the Clinical Practice Research Datalink (CPRD GOLD), linked (person-level) with hospital episode statistics (HES) and death registrations from the Office for National Statistics (approved study protocol number 19_267). Details of the study population have been described in a previous work [12], with the exception of 23 additional individuals identified, after an amendment to the original protocol, with Cockayne and Angelman syndrome; details of these 23 individuals were received in August 2021 (due to COVID-19 delays; please see the data flow diagram in Appendix A for the initial extract and the study population used for the current study). Briefly, the CPRD is an electronic health record primary care research database which is broadly representative of the national population in terms of age, gender and ethnicity [17]. Only GP surgeries in England that consented to their data being linked with hospital episode statistics (HES) and death data (approximately 75% of CPRD surgeries in England) were included in this study.

### 2.2. Sample Population

Initial inclusion criteria for the broader programme of work on which this study was based were: registered at the GP surgery at any point between 1 January 2000 and 29 September 2019 and 10 years old or older to account for delays in reporting of diagnoses of intellectual disability in children [18]. A random sample of 980,586 people without intellectual disabilities (initially 1 million prior to exclusions; see Appendix A) was used for the comparison group with the same eligibility criteria (but without a diagnosis of intellectual disability). For this study, data were further restricted to the 2017–2019 observation period such that people entered the study on 1 January 2017 if this was after the original date of cohort entry or were excluded if they were last seen or died before this date. The final population comprised 7794 individuals with intellectual disabilities and 176,807 individuals without intellectual disabilities (*n* = 440 of whom changed status within the observation window at their first intellectual disability diagnosis).

### 2.3. Definition of Intellectual Disabilities and Health Needs

Diagnostic codes (Read codes and International Classification of Diseases (ICD)-10 codes) for intellectual disabilities and health needs are reported in the Appendix A. These were based on a combination of previous literature [12], free text searching of diagnostic code descriptions and clinical opinion (RK, SKG, RM). The initial choice of health needs was based on the literature in this area [19] and discussions with carers and people living with intellectual disabilities as being sufficiently severe to affect life expectancy. These were: epilepsy; incontinence (urinary or faecal); severe visual loss; severe hearing impairment; severe mobility difficulties; cerebral palsy and feeding via a percutaneous endoscopic gastrostomy (PEG) tube (i.e., as a measure of severe eating/feeding difficulties). To avoid inclusion of shorter-term health needs that had resolved over time and/or been misdiagnosed in childhood, such as epilepsy [20,21], health needs were defined as being present only if their most recent diagnosis was within 10 years of cohort entry. The exception to this was cerebral palsy, which was defined by a diagnosis ever being present given that it is a life-long condition from birth/early infancy [22].

### 2.4. Statistical Methods

The date of entry into the cohort was defined as the latest date according to the person and practice’s characteristics: the beginning (i.e., 1 January 2017) of the observation window; the date of registration with the GP practice; the date the practice was defined as being up to standard (using the CPRD’s own quality indicators); or the date the individual turned 10 years old (to align with the eligibility criteria). Because there are known delays in reporting intellectual disability diagnoses [23] and to avoid conditioning on the future, an intellectual disability status was treated as an age-dependent covariate such that people with intellectual disabilities contributed to the comparison cohort prior to their first diagnosis. Health needs were also treated as age dependent, and individuals contributed to both the presence and absence of health need at different ages if they were diagnosed with a new health need during the observation period. The date of exit was defined as: the date of the last CPRD update (29 September 2019); the date of death; the date of the end of the calendar period; the date of the last practice update or the date of transfer out of practice, whichever was first. The cohort was also sub-divided into individuals without any health needs at baseline or follow-up to assess whether life expectancy was similar between people with and without intellectual disabilities (i.e., excess mortality could be explained by the health needs).

The methodology for the life expectancy work used in this study has been described in detail elsewhere [24]. Life expectancy and 95% confidence intervals (CIs) were compared for people with and without intellectual disabilities and by the presence/absence of each health need using flexible parametric models with intellectual disability and health need status treated as age-varying covariates (and an interaction term fitted). Knots were placed according to the event distribution in the intellectual disability group for greater statistical precision. All models used 5 knots (including the boundary knots; 4 degrees of freedom (df); 3df for age-varying effects) with the exception of PEG feeding, which used 4 knots (3df; 2df for age-varying effects) owing to the small sample size.

## 3. Results

### 3.1. Baseline Characteristics

Table 1 shows the characteristics of the study population over the observation period. The characteristics of the population with each individual health need are shown in Appendix A. In comparison to the rest of the population, people with intellectual disabilities were generally younger (median age 33 vs. 43 years) and more were male (57.1% vs. 49.0%). There were also more white individuals (77.0% vs. 67.5%), although this partly reflects more complete recording of ethnicity in hospital settings (only 14.0% vs. 19.6% had missing data because more people with LD were hospitalised and had their ethnicity recorded). Most individuals (73.4%) with intellectual disabilities had no cause identified: the most common genetic/chromosomal condition reported was Down syndrome (10.9% of the individuals). People with intellectual disabilities had a substantially higher proportion of all of the health needs under investigation compared to those without intellectual disabilities, as is reflected in the greater proportion without any health needs at baseline and follow-up (53.6% vs. 90.3%; intellectual disability vs. no intellectual disability).

The largest differences between people with and without intellectual disabilities were observed for cerebral palsy, which was ≈58 times more prevalent during the 2.7-year observation window (i.e., at baseline or follow-up). Epilepsy, severe visual loss, severe mobility difficulties and PEG feeding were ≈12–22 times more prevalent; and incontinence and severe health impairment were ≈2–4 times more prevalent. The most common severe health need in people with intellectual disabilities was epilepsy, which was present in 18.7% of the individuals at baseline. For people without intellectual disabilities, incontinence was the most common health need, present in 3.8% of the individuals at baseline.

### 3.2. Life Expectancy

Figure 1a–g shows the life expectancy estimates and percentage of life years lost (compared with the general population without health needs), for the severe health needs under investigation, by presence/absence of the health need and intellectual disability status. The final figure (Figure 1h) shows the life expectancy estimates for people without any of the health needs under investigation. Table 2 also presents the exact life expectancy estimates (with 95% CI) at 10, 20 and 40 years old.

Perhaps the most striking finding from the figures is that life expectancy was substantially higher across the board in people with neither intellectual disabilities nor specified health need. At 10 years of age, these individuals could expect to live between 72.2 (absence of severe health impairment) and 74.3 additional years (all of the health needs absent). At the same age, children with intellectual disabilities but without each specified health need lost ≈15–22% of life years compared to this first group, living, on average, an additional 57–62 years. Those with intellectual disabilities but without any of the health needs under investigation needs lost ≈12% of life years, living on average 8.9 years shorter than those with neither intellectual disabilities nor health needs.

We can see that the most severe of the health needs, regardless of intellectual disability status, were PEG feeding and severe mobility difficulties. Ten-year-old children with a PEG feeding tube could expect to live only an additional 23.0 years (95% CI 17.1–31.0) if they had intellectual disabilities and 25.7 years (95% CI 18.3–36.0) if they did not have intellectual disabilities, representing a loss in life years of ≈65–68% compared to those with neither condition. Similarly, children with severe mobility difficulties lost ≈41–44% of life years, living an additional 41–43 years only compared to the almost 73 years in those with neither condition. The disadvantages for individuals with PEG feeding tubes and severe mobility difficulties continued to be observed in adulthood (Table 2).

Of the remaining health needs, people with epilepsy had shorter life expectancy overall. Confidence overlapped at 10 years but, subsequently, having intellectual disability in addition to epilepsy incurred additional life expectancy disadvantages (see Table 2), with a loss in life years of ≈38%. Severe visual loss or incontinence was about equivalent to having intellectual disabilities without the health need in terms of life expectancy, but people with both intellectual disabilities and incontinence were again further disadvantaged, with a loss in life years of ≈34%. Conversely, we did not find an effect of severe hearing impairment on life expectancy. The effect of cerebral palsy on life expectancy was harder to determine, owing to small numbers, but those with cerebral palsy and intellectual disabilities had the shortest life expectancy compared with the those without cerebral palsy or with cerebral palsy but without intellectual disabilities, with a loss in life years of ≈43%. All of these findings were relatively consistent across the age range (Table 2).

## 4. Discussion

This work deepens our understanding of health inequalities in people with intellectual disabilities. By reaffirming that severe health needs make a significant contribution to the mortality disparities that people with intellectual disabilities are known to experience, our findings also reveal that they only partially explain these. After restricting the study population to those without health needs, life expectancy remained shorter for those with intellectual disabilities, with a loss in life years of 12%. Of those with the specified health needs, life expectancy was generally further shortened if intellectual disability was also present, suggesting combined disadvantages.

### 4.1. Strengths and Limitations

To the best of our knowledge, this is the first time that life expectancy has been explored by health needs in people with and without intellectual disabilities. The utilisation of flexible parametric methods to estimate life expectancy is also a novel component and supports previous methodological findings that borrowing strength from larger covariate samples can be an effective way of increasing statistical precision for small samples [24]. However, we recognise that life expectancy is only a crude measure of health inequalities that does not encapsulate other social determinants of health, such as deprivation, or factors that may contribute towards inequalities, such as access to and quality of healthcare provision. We are also unable to comment on other equally important health indicators, including quality of life and well-being.

As with all electronic health record data of this nature which rely on Read code and ICD diagnoses, we are unable to capture variability in the severity of health needs between people with intellectual disabilities and the general population. A particular concern is incontinence, which is likely to be less severe in the general population if it occurred and was resolved during certain life events, such as post-pregnancy [25]; it is noteworthy that almost three-quarters (73.3%) of the people in the general population with incontinence were female, compared with only half (53.2%) in the intellectual disability population (Appendix A). Moreover, both incontinence and PEG feeding may be indicative of additional comorbidities (e.g., frailty or dysphagia) rather than directly causing mortality [26,27]. Our findings nonetheless support their relationship (even if indirect) with life expectancy. Another limitation of GP health record data is that they do not provide complete information on the severity of intellectual disabilities; we are also likely to have missed people with mild intellectual disabilities who do not have significant support needs and may also be more vulnerable to abuse, discrimination and high-risk behaviours. We also recognise that many of these health needs co-occur and that they are more likely to do so if the individual also has intellectual disabilities, as is reflected in the larger proportion of individuals with at least one health need (46% vs. 10%) and high prevalence of co-occurring health needs, particularly for individuals with cerebral palsy (53% of individuals with intellectual disabilities also had epilepsy; 56% had severe mobility difficulties) and PEG feeding tubes (55% had cerebral palsy; 65% epilepsy and 70% severe mobility difficulties) (Appendix A). This descriptive study does not look in more depth at the co-occurring health needs, nor does it adjust for additional clinically relevant comorbidities, such as dementia, or other social determinants of health which all contribute to the mortality disadvantages that people with intellectual disabilities experience [28]. Such issues could be explored further using propensity score methodologies or multiple logistic regression/time-to-event analyses, which are recommended to further develop the work described here.

This study took place before the COVID-19 pandemic, during which people with intellectual disabilities have been adversely affected owing to increased risk of transmission (e.g., through residential homes and community-based support) and increased risk of respiratory deaths [29,30,31]. People with severe health needs have also been disproportionately affected by COVID-19 [32]. In the current climate, the recommendations made here are, therefore, likely to be more relevant.

### 4.2. Comparison with Existing Literature

The prevalence of severe health needs found at baseline in this study is largely similar to that found in previous research carried out in the UK and internationally. The prevalence of epilepsy was 18.7% (vs. 1.1% in the general population), which corresponds with previous population-based studies from England (18.5% vs. 0.7% (matched age/gender/practice population sample) [19]) and Scotland (18.8% vs. 0.8% [10]). The prevalence of incontinence in the intellectual disability population (13.3%) was lower than previous UK estimates using the CPRD (20.5% [19]), which we attribute to the exclusion of ‘H/O incontinence’ and incontinence diagnoses within 10 years of cohort entry for the current study. The baseline prevalence of 3.8% found in the general population is at the lower end of the estimates of international figures of 3–18% for severe incontinence (urinary only) in adult women (about half of this for men) [33], given that many do not seek support from a healthcare provider [34]. The prevalence of PEG feeding (1.9% vs. 0.1%) also falls within the 5-year incidence rate (1.3%) of PEG procedures in England based on 17,000 per year [35].

In our study, prevalence of severe visual impairment among people with intellectual disabilities (13.0%) was lower than previous estimates in the Netherlands for visual impairment and blindness (13.8% and 5.0%, respectively) but, of the latter population, 40.6% were undiagnosed prior to study commencement [36]. We did not find a relationship between severe hearing impairment and life expectancy, which differs from previous (albeit not statistically significant) work in the general population [37].

The prevalence of cerebral palsy we reported here (8.4% vs. 0.1%) can be interpreted using information from the random general population sample. Given that this sample was drawn from 6.2 million individuals (Appendix A) and that 0.5% of the general population sample had intellectual disabilities, and assuming a representative random sample draw, the prevalence of cerebral palsy in our study was approximately 1.42 per 1000 population, which is similar to birth estimates reported of 2.11 per 1000 population [38] conditional on surviving to 10 years. We would also expect there to be 2276 (i.e., 6.2 times as many) people with cerebral palsy in the entire population sample, which would equate to 29% of people with cerebral palsy having intellectual disabilities. This is within the range reported in the literature, which cites 22–40% of individuals with cerebral palsy having cognitive impairment (IQ < 70) [39], although some studies report figures closer to one-half [40]. It is worth emphasising, however, that many—if not most—people with cerebral palsy do not have intellectual disabilities and it is important to make sure that their needs are adequately met. The prevalence of severe mobility difficulties (10.5%) in people with intellectual disabilities is similar to previous estimates of 9.2% for being ‘non-mobile’ [41], but it is hard to determine the prevalence in the general population owing to the recognised variation in thresholds for reporting mobility disabilities [42].

### 4.3. Recommendations

Given that this descriptive study did not seek to control for other contributing factors, we are nonetheless able to make some broad recommendations. First, it is clear from our findings that health needs are a significant problem for people with intellectual disabilities and that, with the exception of severe hearing impairments, they play a key role in shortening life expectancy. More effective management and treatment of these health needs, including regular assessment of associated care requirements, have the potential to improve outcomes and quality of life for those affected. Many of these health needs are relatively rare in the general population, so the development of tailored care pathways for people with intellectual disabilities, based on national guidelines and policies where available, is likely to be a priority. Such pathways may include monitoring medication (e.g., epilepsy—with a focus on epilepsy syndromes and tuberous sclerosis), provision of specialist support (e.g., visual impairment and hearing impairment), communication plans (e.g., cerebral palsy), pain management (e.g., severe mobility difficulties), prevention strategies (e.g., incontinence) and oral care (e.g., PEG feeding). All pathways should include mechanisms for the provision of coordinated care between health, social care and voluntary services so that unnecessary burden is not placed on carers. They should also be adequately flexible to allow for individuals’ differences and needs.

## 5. Conclusions

We conclude that differential life expectancy in people living with intellectual disabilities compared to the general population is not wholly attributable to increased prevalence of severe health needs. Our findings highlight the need to continue to find ways to improve health outcomes and quality of life for people living with intellectual disabilities so that they can be supported to lead long and fulfilling lives.

## Figures and Tables

**Figure 1 ijerph-19-06602-f001:**
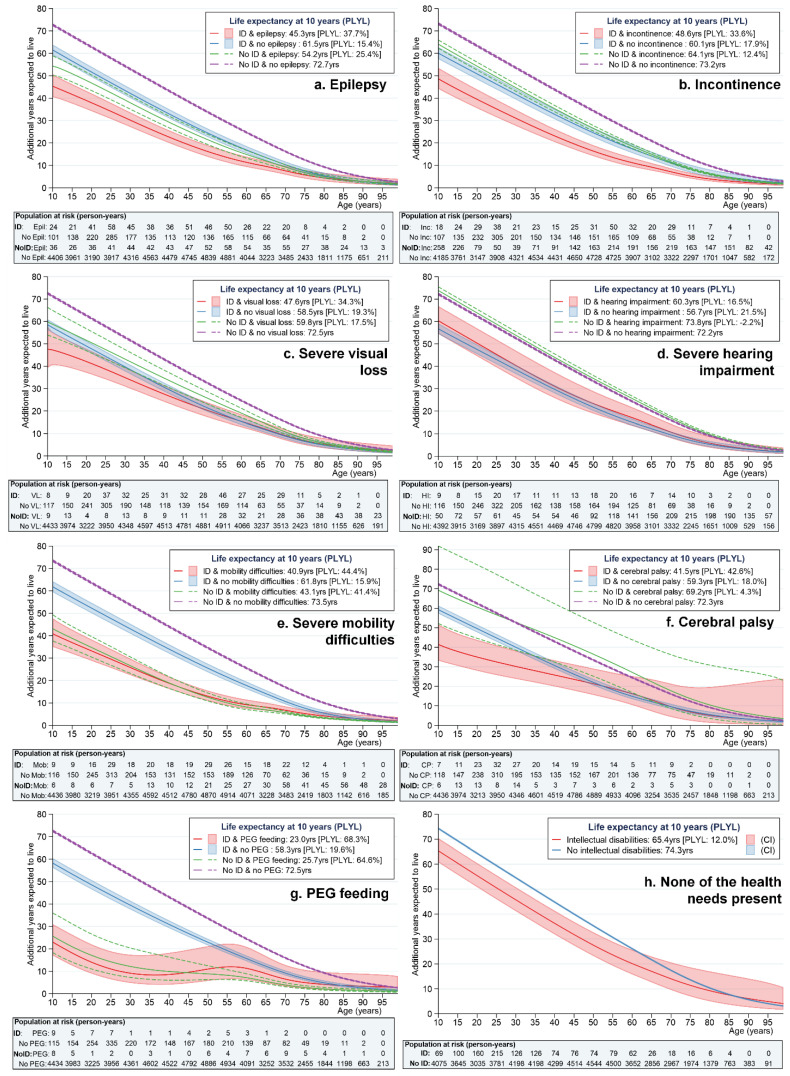
(**a**–**h**) Life expectancy from 10 years of age by presence/absence of intellectual disability and health needs.

**Table 1 ijerph-19-06602-t001:** Baseline and follow-up characteristics of the study population by intellectual disability and health need status.

Characteristic		Intellectual Disability		No Intellectual Disability ^1^	
		Number/ Median	Percent/ Range	Number/ Median	Percent/ Range
Total		7794	100.0	176,807	100.0
**Demographic characteristics**					
Age (years)		33.0	10–101	43.0	10–108
Gender	Male	4448	57.1	86,669	49.0
Female		3346	42.9	90,138	51.0
Ethnicity	White	6002	77.0	119,403	67.5
South Asian		211	2.7	7662	4.3
Black		207	2.7	6236	3.5
Other		280	3.6	8907	5.0
Not known		1094	14.0	34,599	19.6
**Observation period**					
Length in cohort (years)		1.5	>0.0–2.7	1.9	>0.0–2.7
**Most common genetic/chromosomal syndromes ^2^**					
Down syndrome		848	10.9	-	
Fragile X syndrome		151	1.9	-	
Tuberous sclerosis		60	0.8	-	
Edward syndrome		29	0.4	-	
Prader–Willi syndrome		27	0.3	-	
**Severe health needs**					
None (at baseline or follow-up)		4174	53.6	159,716	90.3
Epilepsy	Baseline	1456	18.7	2004	1.1
during follow-up		55	0.7	201	0.1
Incontinence	Baseline	1039	13.3	6649	3.8
during follow-up		214	2.7	1177	0.7
Severe visual loss	Baseline	1015	13.0	1075	0.6
during follow-up		227	2.9	204	0.1
Severe hearing impairment	Baseline	551	7.1	5253	3.0
during follow-up		67	0.9	592	0.3
Severe mobility difficulties	Baseline	818	10.5	1280	0.7
during follow-up		174	2.2	570	0.3
Cerebral palsy	Baseline	658	8.4	261	0.1
during follow-up		20	0.3	6	<0.1
PEG ^3^ feeding	Baseline	132	1.7	180	0.1
during follow-up		20	0.3	54	<0.1

^1^ *n* = 440 individuals moved from no intellectual disability to intellectual disability sample at first diagnosis during observation window. ^2^ *n* = 831 (10.7%) with phenylketonuria (not defined as a specific syndrome for this study). ^3^ PEG: percutaneous endoscopic gastrostomy.

**Table 2 ijerph-19-06602-t002:** Additional years expected to live at age 10, 20 and 40 years by individual health need status and absence of health needs.

	Intellectual Disability				No Intellectual Disability		
	(11,631 Person-Years)				(296,324 Person-Years)		
	Health Need Present		Health Need Absent		Health Need Present		Health Need Absent
Epilepsy: Additional years expected to live (95% CI): % life years lost ^1^							
At 10 years:	45.3 (40.5–50.7)	37.7%	61.5 (59.2–63.9)	15.4%	54.2 (50.0–58.9)	25.4%	72.7 (72.3–73.1)
At 20 years:	38.0 (34.2–42.1)	39.3%	51.7 (49.4–54.0)	17.5%	46.8 (43.4–50.4)	25.4%	62.8 (62.4–63.2)
At 40 years:	22.6 (20.0–25.6)	47.7%	33.0 (31.0–35.1)	23.8%	29.3 (26.8–32.0)	32.3%	43.2 (42.9–43.6)
Incontinence: Additional years expected to live (95% CI): % life years lost ^1^							
At 10 years:	48.6 (44.2–53.5)	33.6%	60.1 (57.5–62.8)	17.9%	64.1 (62.3–65.9)	12.4%	73.2 (72.8–73.7)
At 20 years:	39.5 (35.5–43.9)	37.3%	50.8 (48.4–53.4)	19.6%	54.3 (52.5–56.0)	14.2%	63.3 (62.9–63.8)
At 40 years:	23.2 (20.4–26.3)	47.2%	32.8 (30.6–35.1)	25.2%	35.3 (33.9–36.8)	19.4%	43.8 (43.4–44.2)
Severe visual loss: Additional years expected to live (95% CI): % life years lost ^1^							
At 10 years:	47.6 (39.2–57.9)	34.3%	58.5 (56.2–60.9)	19.3%	59.8 (53.9–66.3)	17.5%	72.5 (72.1–73.0)
At 20 years:	42.1 (37.1–47.8)	32.8%	49.0 (46.8–51.3)	21.5%	51.6 (46.9–56.8)	17.6%	62.7 (62.3–63.0)
At 40 years:	27.6 (24.2–31.5)	36.0%	30.7 (28.8–32.7)	28.9%	34.7 (31.5–38.3)	19.5%	43.1 (42.8–43.5)
Severe hearing impairment: Additional years expected to live (95% CI): % life years lost ^1^							
At 10 years:	60.3 (54.4–66.8)	16.5%	56.7 (54.4–59.1)	21.5%	73.8 (72.2–75.4)	−2.2%	72.2 (71.8–72.6)
At 20 years:	50.3 (44.5–56.9)	19.0%	47.5 (45.4–49.7)	23.5%	63.8 (62.2–65.4)	−2.4%	62.3 (61.9–62.7)
At 40 years:	31.1 (25.8–37.4)	27.5%	29.7 (27.9–31.6)	30.6%	44.0 (42.5–45.5)	−2.7%	42.8 (42.5–43.2)
Severe mobility difficulties: Additional years expected to live (95% CI): % life years lost ^1^							
At 10 years:	40.9 (35.0–47.8)	44.4%	61.8 (59.4–64.3)	15.9%	43.1 (37.7–49.3)	41.4%	73.5 (73.0–73.9)
At 20 years:	33.2 (28.8–38.3)	47.4%	52.3 (50.0–54.7)	17.5%	34.7 (30.5–39.6)	44.9%	63.6 (63.2–64.0)
At 40 years:	18.8 (16.2–21.7)	57.4%	34.0 (32.0–36.1)	22.8%	18.9 (16.5–21.7)	57.0%	44.1 (43.7–44.4)
Cerebral palsy: Additional years expected to live (95% CI): % life years lost ^1^							
At 10 years:	41.5 (33.3–51.7)	42.6%	59.3 (57.2–61.4)	18.0%	69.2 (52.2–91.9)	4.3%	72.3 (71.9–72.7)
At 20 years:	35.0 (28.0–43.8)	44.0%	49.6 (47.6–51.6)	20.3%	60.6 (44.6–82.3)	3.2%	62.5 (62.1–62.8)
At 40 years:	25.8 (19.9–33.3)	40.0%	30.7 (28.9–32.5)	28.6%	44.9 (32.4–62.3)	−4.7%	42.9 (42.6–43.3)
PEG ^2^ feeding: Additional years expected to live (95% CI): % life years lost ^1^							
At 10 years:	23.0 (17.1–31.0)	68.3%	58.3 (56.2–60.5)	19.6%	25.7 (18.3–36.0)	64.6%	72.5 (72.1–72.9)
At 20 years:	14.0 (9.0–22.0)	76.8%	48.6 (46.6–50.7)	22.2%	17.1 (11.0–26.6)	72.0%	62.6 (62.3–63.0)
At 40 years:	8.5 (4.0–18.2)	80.2%	30.9 (29.2–32.7)	28.3%	9.9 (6.0–16.3)	77.0%	43.1 (42.8–43.5)
None of the health needs: Additional years expected to live (95% CI): % life years lost ^1^							
At 10 years:	-		65.5 (60.3–71.1)	12.0%	-		74.3 (73.8–74.7)
At 20 years:	-		55.5 (50.4–61.2)	13.9%	-		64.4 (63.9–64.8)
At 40 years:	-		36.5 (31.5–42.3)	19.0%	-		44.8 (44.4–45.3)

^1^ Percentage of life years lost compared with people with neither intellectual disability nor the specified health need. ^2^ PEG: Percutaneous endoscopic gastrostomy.

## Data Availability

Data for this study were obtained from the Clinical Practice Research Datalink (CPRD), provided by the UK MRHA. The authors’ licence for using these data does not allow sharing of raw data with third parties. Information about access to CPRD data is available here: https://www.cprd.com/research-applications (accessed on 20 May 2022). Researchers should contact the ISAC Secretariat at isac@cprd.com for further details.

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
