# Peer review of "Health Needs and Their Relationship with Life Expectancy in People with and without Intellectual Disabilities in England"

_ijerph, 2022, doi:10.3390/ijerph19116602_

Round 1

Reviewer 1 Report

Person-first language uses the terms intellectual and physical challenges rather than disability. However, the authors use non-biased language when they refer to people "living with disabilities". 

Although addressed in limiations, the authors might address how or why they definded variables as "severe health needs". Were these determined by the ICD 10 diagnostic codes? Also, were different health needs weighted differently in the model?

Author Response

We thank the reviewer for these comments.

We have changed a number of the references in the text to people “living with” intellectual disabilities, as opposed to “with intellectual disabilities” as we agree with the reviewer that this is a preferred term – please see tracked changes in the document.

We have also now clarified more fully that the health needs were identified from the ICD and Read codes (see beginning of section 2.3). The choice of health needs were from the literature but also from discussions with family carers and people with intellectual disabilities as being sufficient severe to affect life expectancy (now clarified further in this section). The health needs were investigated individually (albeit adjusted for age, by design) so we did not weight them differently.

Reviewer 2 Report

A very modern and interesting topic. Definitely of interest for the readers. Well written and organized. Statistics part look ok to me, although I am not exactly a big specialist in this area and further expertise should be asked here, besides mine. Introduction sets the scene well. Discussion sections and Conclusions are very well balanced. Paper is easy to read and has a clear relevance in this area and for the present journal. I would recommend the acceptance of it. 

Author Response

We thank the reviewer for these comments

Reviewer 3 Report

As far as I can see, this is a well conducted and well presented study.  

Author Response

We thank the reviewer for this comment